# Connected Hidden Neurons (CHNNet): An Artificial Neural Network for Rapid Convergence

## Abstract

Despite artificial neural networks being inspired by the functionalities of biological neural networks, unlike biological neural networks, conventional artificial neural networks are often structured hierarchically, which can impede the flow of information between neurons as the neurons in the same layer have no connections between them. Hence, we propose a more robust model of artificial neural networks where the hidden neurons, residing in the same hidden layer, are interconnected that leads to rapid convergence. With the experimental study of our proposed model in deep networks, we demonstrate that the model results in a noticeable increase in convergence rate compared to the conventional feed-forward neural network.

## 1 Introduction

The biological neural networks process large amounts of data passed by senses from different parts of the body (Palm, 1986). A brain can have approximately 100 billion neurons and 100 trillion neural connections, which implies that each neuron can have connections with 1000 other neurons (Glasser et al., 2016). Moreover, the neurons in the brain form complex and dense connections among themselves, which is important for efficient and flexible information processing (Sporns, 2013). Although the operation of biological neurons served as inspiration for neural networks as they are used in computers, many of the designs have since gotten very disconnected from biological reality. (Akomolafe, 2013). Artificial neural networks (ANNs) often follow hierarchical structures with simple neural connections that can impede the flow of information between neurons, as the neurons in the same layer have no connections between them. In some scenarios, to improve the generalization power of new and unseen data, it is important to have more connections among the neurons, as a network with more connections can learn more robust and meaningful features (Zhang et al., 2016). Moreover, having more connections among the neurons can potentially speed up the convergence rate, as it helps to learn complex patterns and relations in the data (Goodfellow et al., 2016). We hypothesize that designing a neural network model with an increased number of neural connections will result in a performance gain in terms of learning. In conventional ANNs, specifically in feed-forward neural networks (FNNs), to increase the number of connections while keeping the number of layers fixed, the number of neurons per hidden layer has to be increased (Goodfellow et al., 2016). However, increasing the number of neurons can lead to a slow convergence problem in the model (Gron, 2017). To achieve rapid learning, extensive research has been conducted on various aspects of neural network design, e.g. adaptive gradient methods such as the Adam optimizer (Kingma & Ba, 2014), and activation functions such as the rectified linear unit (ReLU) (Glorot et al., 2010). With a particular focus on the architectural elements that can be adjusted to achieve rapid learning, we propose to connect the hidden neurons of the networks in order to increase the number of neural connections in the network. We propose that the model has the potential to achieve rapid convergence compared to the conventional FNNs while applying the same training strategies. However, connecting all the hidden neurons in a network is compute-intensive, and thus we design an ANN model where the hidden neurons, residing in the same hidden layer, are interconnected, which preserves the parallel computability property of the model as well.

## 1.1 RESEARCH CONTRIBUTION

The primary contributions of the paper are summarized as follows:

- We introduced a neural network model, namely CHNNet (Connected Hidden Neurons), in which we created connections among the hidden neurons residing in the same hidden layer, enabling robust information sharing among the neurons.

- We formulated mathematical equations to calculate the activations of the hidden layers in forward propagation and revised the backpropagation algorithm to calculate the gradients based on the formulated forward propagation equations. Moreover, We provided proof of our claim of rapid convergence.

- The proposed model is different from conventional RNNs in calculating the input from the hidden neurons and is not architecturally equivalent to two conventional FNN layers connected to the previous layer through skip connections.

- We tested the proposed model on benchmark datasets and demonstrated that the model depicted a noticeable increase in convergence rate compared to the conventional FNN model.

- As our model generates a larger number of parameters compared to the conventional FNN model, we tested the proposed model against the FNN model with an increased number of parameters and showed that the model outperformed the FNN model in the mentioned configuration as well.

## 2 LITERATURE REVIEW

In the infancy of neural networks, Minsky & Papert (1969) specified significant drawbacks of perceptrons and suggested the raw idea of Multilayer Perceptron (MLP). The architecture they proposed is hierarchical in structure and has no mention of connections among hidden neurons, residing in the same layer. Further, a few FNN architectures were analyzed in the literature by Rumelhart et al. (1986), none of which featured connections among the hidden neurons of the same layer.

Thus far, a number of ANNs have been introduced using different approaches to establish connections among neurons. A Recurrent Neural Network (RNN) has self-connections among hidden neurons through time; that is, the self-connections work as information carriers from one time step to another (Rumelhart et al., 1986). The Hopfield Neural Network, a single-layered neural network introduced by Hopfield (1982), has neurons symmetrically connected to all other neurons through bidirectional connections. While Hopfield use positive feedback to stabilize the network output, Achler (2014) proposed using negative feedback which regulate the inputs during recognition phase. Similar to the Hopfield Network, the Boltzmann Machine has its neurons connected symmetrically with all other neurons, with the exception that the neurons are divided into visible units and hidden units (Hinton & Sejnowski, 1986). Neural networks like the Echo State Network (ESN) (Jaeger, 2001) and Liquid State Machine (LSM) (Maass et al., 2002) have featured a pool of neurons, namely a reservoir, which consists of numerous randomly connected neurons, providing them with non-linear modeling ability. However, as the reservoir is randomized, it requires numerous trials and sometimes even luck (Ozturk et al., 2007). Additionally, in Spiking Neural Networks (SNNs), recurrent connections (Zhang et al., 2021) and self-connections (Zhang & Zhou, 2022) in the hidden layer have been proposed, which require a spiking version of the actual input data to be implemented. The referred ANNs have recurrent connections among the neurons that are different from the proposed connections among the hidden neurons of our model.

In the contemporary period, designing new paths for information flow in neural networks has attained noticeable success. Convolutional Neural Network (CNN) architectures like DenseNet (Huang et al., 2017), ResNet (He et al., 2016), and UNet++(Zhou et al., 2018), which use skip connections to directly pass information from a layer to a deeper layer, have reached state-of-the-art (SOTA) performance. Moreover, Liu et al. (2022) have introduced the Group Neural Network, which, to overcome the blockade at information passing, features a group of neurons that can connect freely with each other. However, due to its irregular architecture, the training of the network cannot be accelerated through parallel computing. The mentioned ANNs use different approaches to enable information flow among the hidden neurons than ours.

## 3 METHODOLOGY

The proposed architecture features additional self-connections and interconnections among the hidden neurons, as shown in figure 1.

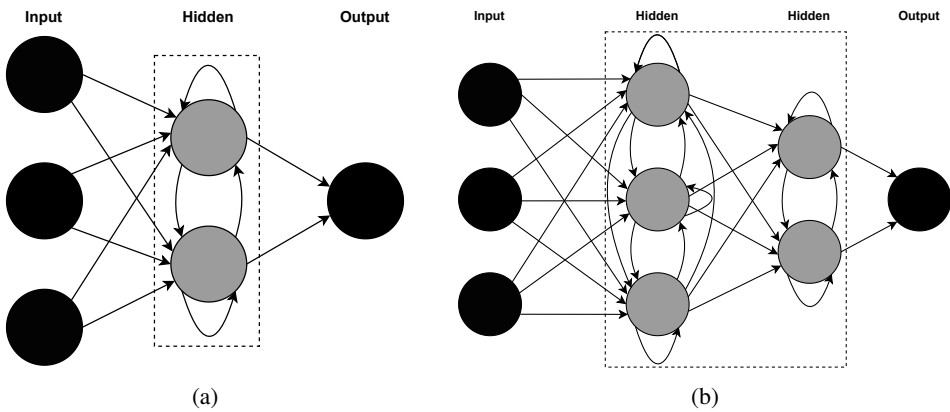

Figure 1: Proposed architecture of CHNNet with (a) one hidden layer and (b) two hidden layers.

We have formulated mathematical equations for forward propagation and revised the backpropagation algorithm for hidden layers only, as no new connections have been introduced in the input and output layers.

### 3.1 FORWARD PROPAGATION

Rumelhart et al. (1986) note that the process of calculating the activations of each layer in the forward direction is straightforward and can be done quickly using matrix operations. Mathematically, forward propagation can be expressed as follows:

Let $f$ be the activation function. Then, for the $l^{th}$ hidden layer, the input is $\boldsymbol{A}^{[l-1]}$ and the output $\boldsymbol{A}^{[l]}$ is computed as:

$$\boldsymbol{Z}^{[l]} = \boldsymbol{W}^{[l]}\boldsymbol{A}^{[l-1]} + \boldsymbol{B}^{[l]}$$
$$\boldsymbol{A}^{[l]} = f(\boldsymbol{Z}^{[l]})$$

where $\boldsymbol{W}^{[l]}$ is the weight matrix connecting $(l-1)^{th}$ layer to $l^{th}$ layer, $\boldsymbol{B}^{[l]}$ is the bias matrix of $l^{th}$ layer, and $\boldsymbol{Z}^{[l]}$ and $\boldsymbol{A}^{[l]}$ are the pre-activation and post-activation of $l^{th}$ layer respectively.

Unlike the conventional FNN architecture, in CHNNet, information from one hidden neuron is consolidated into other hidden neurons residing in the same hidden layer. Therefore, for the forward propagation, we have two sets of weight matrices, one connecting the $(l-1)^{th}$ layer to $l^{th}$ layer and the other connecting hidden neurons of the $l^{th}$ layer to other hidden neurons of the layer. Then for layer $l$, the input is $\boldsymbol{A}^{[l-1]}$ and the pre-activation $\boldsymbol{Z}^{[l]}$ is proposed to be computed as:

$$\boldsymbol{Z}^{[l]} = \boldsymbol{W}_1^{[l]}\boldsymbol{A}^{[l-1]} + \boldsymbol{W}_2^{[l]}\boldsymbol{H}^{[l]} + \boldsymbol{B}^{[l]} \tag{1}$$

where $\boldsymbol{W}_1^{[l]}$ is the weight matrix connecting $(l-1)^{th}$ layer to $l^{th}$ layer, $\boldsymbol{W}_2^{[l]}$ is the weight matrix connecting hidden neurons of $l^{th}$ layer to other hidden neurons of the layer, $\boldsymbol{B}^{[l]}$ is the bias matrix $l^{th}$ layer, $\boldsymbol{H}^{[l]}$ is the input from the hidden neurons of $l^{th}$ layer, and $\boldsymbol{Z}^{[l]}$ and $\boldsymbol{A}^{[l]}$ are the pre-activation and post-activation of $l^{th}$ layer respectively.

The input from the hidden neurons, $\boldsymbol{H}^{[l]}$ in equation 1, is the new term introduced in the conventional forward propagation equation. As yet, there are not many mechanisms available to calculate the output of the hidden neurons given an input. In the proposed model, the pre-activation of the $l^{th}$ hidden layer is used to calculate $\boldsymbol{H}^{[l]}$. Thereby, for $l^{th}$ layer, the input is $\boldsymbol{A}^{[l-1]}$ and the input from the hidden neurons $\boldsymbol{H}^{[l]}$ is computed as:

$$\boldsymbol{H}^{[l]} = \boldsymbol{W}_1^{[l]}\boldsymbol{A}^{[l-1]} + \boldsymbol{B}^{[l]} \tag{2}$$

Finally, the post-activation $\boldsymbol{A}^{[l]}$ of $l^{th}$ layer is computed as:

$$\boldsymbol{A}^{[l]} = f(\boldsymbol{Z}^{[l]})$$

Though the forward propagation mechanism of the proposed model echoes the forward propagation mechanism of conventional RNNs, in conventional RNNs, the activations of the hidden neurons, obtained from the prior time step, are used to calculate the output of the hidden layer, whereas in CHNNet, the current pre-activations of the hidden neurons are used to calculate the output of the hidden layer. Moreover, there is an argument to be made that the choice of mechanism to calculate the input from the hidden neurons, $\boldsymbol{H}^{[l]}$, has conceptually led to generating two conventional FNN layers connected to the previous layer through skip connections. However, any non-linear activation function has not been used to calculate the value of $\boldsymbol{H}^{[l]}$. Thereby, as artificial neurons need to be associated with a non-linear activation function (Minsky & Papert, 1969), $\boldsymbol{H}^{[l]}$ cannot be considered as the input from an individual layer of neurons.

## 3.2 BACKPROPAGATION

As described by Rumelhart et al. (1986), the backpropagation can be mathematically expressed as follows:

Let $\boldsymbol{Y}$ be the output, $f$ be the activation function, and $\boldsymbol{E}$ be the cost. The upstream gradient for the output layer $\boldsymbol{D}_u^{[L]}$, where $L$ is the number of layers in the network, is computed as:

$$\boldsymbol{D}_u^{[L]} = \nabla_{\boldsymbol{Y}}\boldsymbol{E}$$

Let $\nabla_{\boldsymbol{Z}^{[l]}}\boldsymbol{E} = \boldsymbol{D}^{[l]}$, where, $\boldsymbol{Z}^{[l]}$ be the pre-activation of $l^{th}$ layer. For $l^{th}$ layer, $\boldsymbol{D}^{[l]}$ is computed as:

$$\boldsymbol{D}^{[l]} = \boldsymbol{D}_u^{[l]} \odot f'(\boldsymbol{Z}^{[l]})$$

Then, the partial derivatives of the cost with respect to weight matrix $\boldsymbol{W}^{[l]}$ and bias matrix $\boldsymbol{B}^{[l]}$ are computed as:

$$\nabla_{\boldsymbol{W}^{[l]}}\boldsymbol{E} = \boldsymbol{D}^{[l]}\boldsymbol{A}^{[l-1]^{\mathsf{T}}}$$

$$\nabla_{\boldsymbol{B}^{[l]}}\boldsymbol{E} = \boldsymbol{D}^{[l]}$$

where $\boldsymbol{A}^{[l-1]}$ is the input from $(l-1)^{th}$ layer. Finally, the weight matrix $\boldsymbol{W}^{[l]}$ and bias matrix $\boldsymbol{B}^{[l]}$ are updated using the following equations:

$$\boldsymbol{W}^{[l]} \rightarrow \boldsymbol{W}^{[l]} - \eta\nabla_{\boldsymbol{W}^{[l]}}\boldsymbol{E} \tag{3}$$

$$\boldsymbol{B}^{[l]} \rightarrow \boldsymbol{B}^{[l]} - \eta\nabla_{\boldsymbol{B}^{[l]}}\boldsymbol{E} \tag{4}$$

where $\eta$ is the learning rate of the network.

In contrast to conventional FNN architectures, CHNNet has two sets of weight matrices. The weight matrix $\boldsymbol{W}_1^{[l]}$ and bias matrix $\boldsymbol{B}^{[l]}$ are updated using equation 3 and equation 4 respectively. The partial derivative of the cost with respect to weight matrix $\boldsymbol{W}_2^{[l]}$ is computed as:

$$\nabla_{\boldsymbol{W}_2^{[l]}}\boldsymbol{E} = \boldsymbol{D}^{[l]}\boldsymbol{H}^{[l]^{\mathsf{T}}} \tag{5}$$

Then, the weight matrix $\boldsymbol{W}_2^{[l]}$ is updated using the following equation:

$$\boldsymbol{W}_2^{[l]} \rightarrow \boldsymbol{W}_2^{[l]} - \eta\nabla_{\boldsymbol{W}_2^{[l]}}\boldsymbol{E} \tag{6}$$

In the end, for $(l-1)^{th}$ layer, the upstream gradient $\boldsymbol{D}_u^{[l-1]}$ is computed as:

$$\boldsymbol{D}_u^{[l-1]} = \boldsymbol{D}^{[l]}\boldsymbol{W}_1^{[l]^{\mathsf{T}}}$$

### 3.3 PROOF OF RAPID CONVERGENCE

For the purpose of proving the claim of rapid convergence, let, at time step $t$, $C_C(w_1^t, w_2^t)$ be the cost of a hidden layer of CHNNet with weights $w_1^t$ and $w_2^t$ and $F_C(w_1^t)$ be the cost of a hidden layer of the conventional FNN with weight $w_1^t$. Mathematically, the cost function $C_C(w_1^t, w_2^t)$ and $F_C(w_1^t)$ can be expressed as:

$$C_C(w_1^t, w_2^t) = ||O^* - f(C_F(w_1^t, w_2^t))||$$
$$F_C(w_1^t) = ||O^* - f(F_F(w_1^t))||$$

where, $C_F(w_1^t, w_2^t)$ is the pre-activation of the hidden layer of CHNNet with weights $w_1^t$ and $w_2^t$ at time step $t$, $F_F(w_1^t)$ is the pre-activation of the hidden layer of the conventional FNN with weight $w_1^t$ at time step $t$, $f$ is the activation function and $O^*$ is the optimal output of the hidden layer.

Let, at time step $t$, $m^t = F_F(w_1^t) = w_1^t a^t + b^t$ and $c^t = w_2^t h^t$ where, $a^t$ is the activation of the previous layer and $h^t$ is the input from the hidden neurons of the current layer. Thus, we have,

$$C_F(w_1^t, w_2^t) = m^t + c^t \tag{7}$$

Here, $h^t = m^t$, and thus we get from equation 7,

$$C_F(w_1^t, w_2^t) = m^t + w_2^t m^t$$
$$\Rightarrow C_F(w_1^t, w_2^t) = (1 + w_2^t) F_F(w_1^t) \tag{8}$$

When $w_2^{t-1} = 0$ at time step $t - 1$, we get using equation 8,

$$||O^* - f(C_F(w_1^{t-1}, w_2^{t-1}))|| = ||O^* - f(F_F(w_1^{t-1}))|| \Rightarrow C_C(w_1^{t-1}, w_2^{t-1}) = F_C(w_1^{t-1}) \tag{9}$$

Then, as gradient descend is guaranteed to converge according to the convergence theorem of gradient descend, $w_2$ is updated such that $f(C_F(w_1, w_2)) \rightarrow O^*$. Therefore, at time step $t$ we get,

$$||O^* - f(C_F(w_1^t, w_2^t))|| \leq ||O^* - f(F_F(w_1^t))|| \Rightarrow C_C(w_1^t, w_2^t) \leq F_C(w_1^t) \tag{10}$$

Using equation 9 and inequality 10, we get,

$$C_C(w_1^{t-1}, w_2^{t-1}) - C_C(w_1^t, w_2^t) \geq F_C(w_1^{t-1}) - F_C(w_1^t) \tag{11}$$

Equation 11 implies that the difference between the cost of CHNNet, generated at two sequential time steps, is greater than that of the conventional FNN; that is, CHNNet converges faster than the conventional FNN.

## 4 PERFORMANCE EVALUATION

To evaluate the performance of the proposed model, we used the software library TensorFlow. Using the library, we constructed a layer, namely the CHN Layer, implementing the forward propagation and backpropagation mechanisms described in section 3. Using the layer, along with other layers provided by the TensorFlow library, we performed all our experiments. Our goal was not to get SOTA performance on the benchmark datasets. Rather, it was to achieve a better convergence rate than the conventional FNN. We compared the performance of the CHN layers with that of the Dense layers provided by the TensorFlow library, which implement conventional forward propagation and backpropagation mechanisms of FNN. In our initial experiments, CHNNet generated a larger number of parameters compared to FNN, and thus we conducted some additional experiments with CHNNet and FNN having nearly equal numbers of parameters.

### 4.1 DATASETS

We evaluated the performance of the CHNNet on three benchmark datasets of different sizes and diverse features, namely the MNIST (LeCun et al., 2010), the Fashion MNIST (Xiao et al., 2017), and the Extended MNIST (Cohen et al., 2017) datasets. The MNIST dataset, consisting of 60,000 training samples and 10,000 testing samples, holds 28x28 images of handwritten digits divided into 10 classes. The Fashion MNIST (FMNIST) dataset has the same features as the MNIST dataset, with the exception that the dataset contains 10 classes of fashion accessories instead of handwritten digits. In addition, the FMNIST dataset is more complex compared to the MNIST dataset. The Extended MNIST (EMNIST) dataset, consisting of 697,932 training samples and 116,323 testing samples, contains 28x28 images of handwritten digits and letters divided into 62 classes. The EMNIST dataset is profoundly more complex than both the MNIST and FMNIST datasets.

## 4.2 Hyperparameters

We chose three different architectures that vary in terms of the number of hidden neurons each and the total number of layers for each of the datasets, and conducted each experiment with three seeds. The loss functions for the experiments were chosen depending on the type of dataset and the desired output format. Moreover, we used categorical cross entropy in multi-class classification problems (Goodfellow et al., 2016). The optimizers and learning rates are selected based on the improvement they could bring to the performance of the models.

## 4.3 Experiments on deep networks

### 4.3.1 Training parameters

For the MNIST dataset, we used networks consisting of 4 hidden layers with 96 neurons each, 6 hidden layers with 256 neurons each, and a network as "288-256-224-192-160-128-96-64-10". Additionally, We used the RMSprop optimizer with a learning rate of 0.0001, sparse categorical cross entropy as the loss function and batches of size 512. We used networks consisting of 3 hidden layers with 512 neurons each, 6 hidden layers with 256 neurons each, and a network as "928-800-672-544-416-288-160-32-10" for training on the FMNIST dataset. Moreover, we used the SGD optimizer with a learning rate of 0.001, sparse categorical cross entropy as the loss function and batches of size 32 for the training. While training on the EMNIST dataset, we used networks consisting of 3 hidden layers with 768 neurons each, 6 hidden layers with 320 neurons each, and a network as "1024-896-768-640-512-348-256-128-62". We used the SGD optimizer with a learning rate of 0.001, sparse categorical cross entropy as the loss function and batches of size 32 for training the models. The networks had ReLU activation in the hidden layers and softmax activation in the output layer. Further, we performed t-tests on the sets of accuracies achieved by the conventional FNN and CHNNet through the networks and obtained the p-values and t-statistics. A small p-value indicates that the mean accuracies of FNN and CHNNet are not identical. Furthermore, smaller p-values are associated with larger t-statistics.

### 4.3.2 Test results

The CHNNet showed a considerable performance gain in terms of convergence compared to the conventional FNN with all the architectures, as portrayed in figure 2. In addition, CHNNet showed a better performance, on average, than the conventional FNN in terms of mean loss and mean accuracy, as shown in table 1 and table 2. Moreover, in terms of accuracy, the experiments depicted negative t-statistics with all the architectures, which suggests that CHNNet had a higher mean accuracy than the conventional FNN in the experiments.

Table 1: Loss measurement of CHNNet and FNN in deep networks.

| Datasets | Model | FNN | | CHNNet | |
| --- | --- | --- | --- | --- | --- |
| | | Trainable Params | Mean Loss (±std) | Trainable Params | Mean Loss (±std) |
| MNIST | Arch-1 | 104,266 | 0.187(±0.004) | 141,130 | 0.142 (±0.003) |
| | Arch-2 | 532,490 | 0.125(±0.013) | 925,706 | 0.110(±0.010) |
| | Arch-3 | 471,562 | 0.153(±0.008) | 762,378 | 0.138(±0.027) |
| FMNIST | Arch-1 | 932,362 | 0.351(±0.004) | 1,718,794 | 0.319(±0.003) |
| | Arch-2 | 532,490 | 0.339(±0.004) | 925,706 | 0.336(±0.014) |
| | Arch-3 | 2,774,602 | 0.334(±0.004) | 5,305,930 | 0.342(±0.013) |
| EMNIST | Arch-1 | 1,831,742 | 0.429(±0.001) | 3,601,214 | 0.411(±0.004) |
| | Arch-2 | 784,702 | 0.429(±0.001) | 1,193,982 | 0.426(±0.001) |
| | Arch-3 | 3,567,934 | 0.422(±0.005) | 6,910,270 | 0.419(±0.004) |

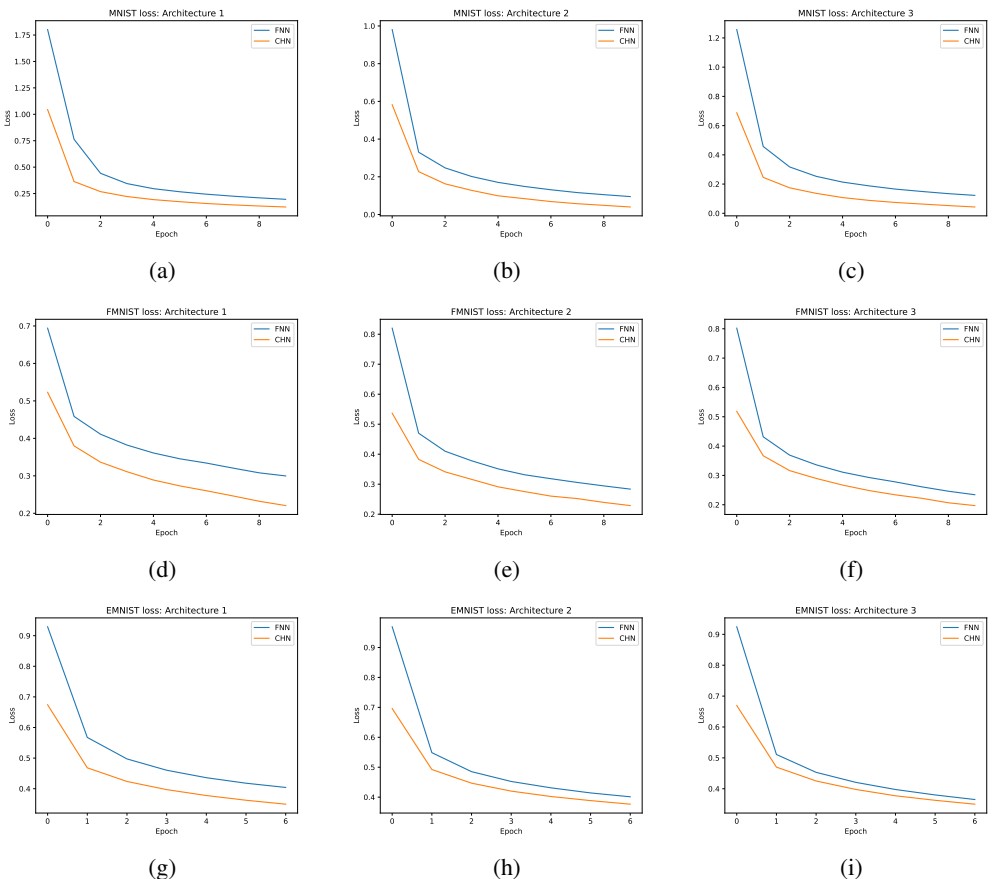

Figure 2: Loss curves of three different architectures of CHNNet and FNN on the (a)-(c) MNIST, (d)-(f) FMNIST, and (g)-(i) EMNIST datasets.

Table 2: Accuracy measurement of CHNNet and FNN in deep networks.

| Datasets | Model | FNN Mean Accuracy (±std) | CHNNet Mean Accuracy (±std) | p-value | t-statistics |
|---|---|---|---|---|---|
| MNIST | Arch-1 | 94.47(±0.17) | 95.62(±0.13) | 0.03 | -6.20 |
| | Arch-2 | 96.13(±0.46) | 96.71 (±0.30) | 0.13 | -2.47 |
| | Arch-3 | 95.41(±0.28) | 96.16(±0.75) | 0.41 | -1.02 |
| FMNIST | Arch-1 | 87.35(±0.24) | 88.68 (±0.11) | 0.03 | -5.51 |
| | Arch-2 | 87.71(±0.10) | 88.11(±0.20) | 0.10 | -2.93 |
| | Arch-3 | 88.13(±0.32) | 88.56(±0.30) | 0.29 | -1.42 |
| EMNIST | Arch-1 | 84.76(±0.02) | 85.25(±0.05) | 0.004 | -14.97 |
| | Arch-2 | 84.67(±0.07) | 84.73(±0.02) | 0.46 | -0.91 |
| | Arch-3 | 84.89 (±0.21) | 85.16(±0.09) | 0.15 | -2.29 |

## 4.4 EXPERIMENTS WITH EQUAL PARAMETERS

In the experiments conducted previously, the CHNNet generated more parameters compared to the conventional FNN. Hence, we conducted additional experiments with an increased number of hidden neurons in the Dense layers of FNN to evaluate the performance of CHNNet compared to FNN with a nearly equal number of parameters.

### 4.4.1 TRAINING PARAMETERS

The training parameters were the same as in previous experiments, except that we increased the number of hidden neurons in the Dense layers. Hence, for the conventional FNN, we used an architecture which feature 4 hidden layers with 126 neurons in each layer, 6 hidden layers with 360 neurons each, and 8 hidden layers with "360-334-304-268-238-208-176-142" neurons in the respective hidden layers for the MNIST dataset. For the FMNIST dataset, we used architectures with 3 hidden layers with 749 neurons each, 6 hidden layers with 358 neurons each, 8 hidden layers with "1184-1056-928-800-704-604-448-352" neurons in the respective hidden layers. For the EMNIST dataset, architectures featured 3 hidden layers with 1152 neurons each, 6 hidden layers with 412 neurons each, and 8 hidden layers with "1272-1144-1016-978-760-632-504-376" neurons in the respective hidden layers.

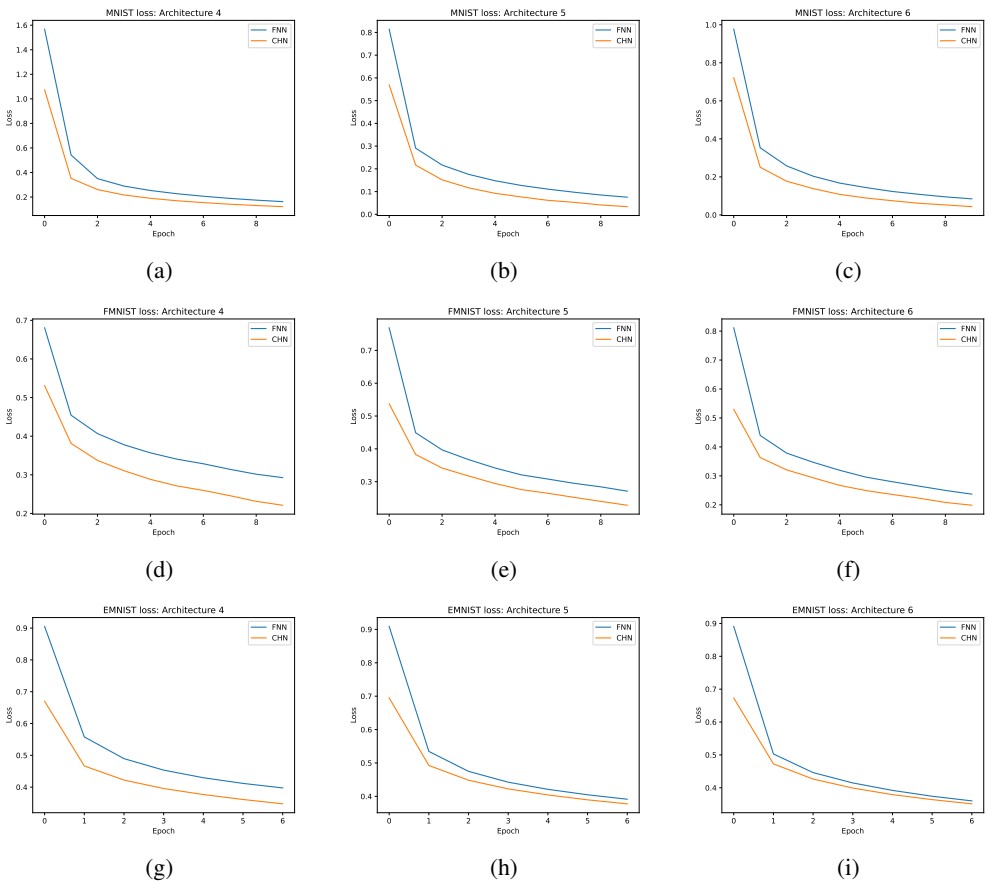

Figure 3: Loss curves of three different architectures of CHNNet and FNN on the (a)-(c) MNIST, (d)-(f) FMNIST, and (g)-(i) EMNIST datasets with nearly equal numbers of parameters.

### 4.4.2 TEST RESULTS

Despite increasing the number of neurons in the hidden layers of the conventional FNN, CHN-Net showed a considerably faster convergence rate with all the architecture, as depicted in figure 3. Moreover, the CHNNet commonly performed better in terms of mean loss and mean accuracy compared to the conventional FNN, as illustrated in tables 3 and 4. The experiments depicted negative t-statistics with the architectures except for one architecture on both the MNIST and FMNIST datasets and two architectures on the EMNIST dataset, which is interpreted as CHNNet commonly outperforming FNN in terms of mean loss and mean accuracy. Further, it can be concluded that the larger number of parameters generated by CHNNet is not a matter of concern, as even with a

nearly equal number of parameters in both CHNNet and the conventional FNN model, CHNNet outperformed the FNN model in terms of convergence.

Table 3: Loss measurement of CHNNet and FNN with nearly equal number of parameters.

| Datasets | Model | FNN | | CHNNet | |
|---|---|---|---|---|---|
| | | Trainable Params | Mean Loss (±std) | Trainable Params | Mean Loss (±std) |
| MNIST | Arch-4 | 148,186 | 0.164(±0.002) | 141,130 | 0.151(±0.004) |
| | Arch-5 | 936,010 | 0.115(±0.015) | 925,706 | 0.111(±0.006) |
| | Arch-6 | 763,836 | 0.146(±0.035) | 762,378 | 0.228(±0.085) |
| FMNIST | Arch-4 | 1,718,965 | 0.345(±0.004) | 1,718,794 | 0.320(±0.003) |
| | Arch-5 | 927,230 | 0.339(±0.006) | 925,706 | 0.336(±0.008) |
| | Arch-6 | 5,327,238 | 0.327(±0.006) | 5,305,930 | 0.341(±0.015) |
| EMNIST | Arch-4 | 3,632,318 | 0.426(±0.004) | 3,601,214 | 0.411(±0.001) |
| | Arch-5 | 1,199,806 | 0.425(±0.004) | 1,193,982 | 0.430(±0.002) |
| | Arch-6 | 6,370,056 | 0.411(±0.003) | 6,369,086 | 0.418(±0.001) |

Table 4: Accuracy measurement of CHNNet and FNN with nearly equal numbers of parameters.

| Datasets | Model | FNN Mean Accuracy (±std) | CHNNet Mean Accuracy (±std) | p-value | t-statistics |
|---|---|---|---|---|---|
| MNIST | Arch-4 | 95.12(±0.08) | 95.39(±0.18) | 0.10 | -2.92 |
| | Arch-5 | 96.59(±0.35) | 96.80(±0.13) | 0.46 | –0.92 |
| | Arch-6 | 95.57(±1.04) | 94.02(±2.16) | 0.21 | 1.83 |
| FMNIST | Arch-4 | 87.52(±0.19) | 88.64(±0.23) | 0.004 | -15.83 |
| | Arch-5 | 87.78(±0.02) | 88.37(±0.10) | 0.02 | -7.37 |
| | Arch-6 | 88.45(±0.27) | 88.43(±0.31) | 0.96 | 0.05 |
| EMNIST | Arch-4 | 84.84(±0.14) | 85.20(±0.06) | 0.03 | -5.83 |
| | Arch-5 | 84.71(±0.11) | 84.66(±0.09) | 0.10 | 2.90 |
| | Arch-6 | 85.18(±0.10) | 85.07(±0.07) | 0.15 | 2.33 |

## 5 CONCLUSION

We designed an ANN, namely CHNNet, that is different from the existing neural networks in connecting the hidden neurons of the same layer. In addition, we described the forward propagation and backpropagation mechanisms of the proposed model and provided proof of the claim of rapid convergence. In the experiments we conducted, CHNNet showed a noticeable increase in convergence rate compared to the conventional FNN model without compromising loss or accuracy. However, the proposed model generated a larger number of parameters compared to the conventional FNN. Thus, we conducted additional experiments and concluded that the larger number of parameters generated by CHNNet is not a matter of concern, as even with a nearly equal number of parameters in both CHNNet and the conventional FNN, CHNNet outperformed FNN in terms of convergence without any compromises in mean loss or mean accuracy.

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
