# OpenReview forum: "Connected Hidden Neurons (CHNNet): An Artificial Neural Network for Rapid Convergence"
_ICLR.cc/2024/Conference — Submitted to ICLR 2024_

### Official Review · Reviewer_fGCE · 2023-10-16

**Soundness:** 1 poor
**Presentation:** 2 fair
**Contribution:** 2 fair
**Rating:** 3
**Confidence:** 5

**Summary:**

The paper proposes a novel MLP architecture where individual hidden neurons in a layer are densely interconnected. The authors derive the back propagation algorithm for the proposed architecture (in broad strokes), and show (theoretically) that the proposed model is expected to have steeper gradients, which the authors associate with faster convergence. Experiments are conducted on MNIST dataset variations, where the proposed model has a favourable training profile.

**Strengths:**

**Originality:** The paper proposes an MLP architecture with self- and neighbourhood-connections within a layer. According to the literature review done by the authors, this is the first instance of such an architecture being proposed. Authors do compare their model to classic RNNs, and I would have appreciated a deeper discussion on the parallels. Either way, I am convinced that the idea is novel.

**Quality:** The authors give some theoretical backing to the rapid convergence claims, which is appreciated.

**Clarity:** The paper is clearly written and very easy to follow.

**Significance:** I believe a broader benchmarking experiment is needed to determine the significance. However, it is nice to see a simple and elegant idea proposed.

**Weaknesses:**

**Benchmarking experiments:** The authors have chosen MNIST and its variations for benchmarking. The choice seems very unobvious: these are image datasets, with lots of “redundant” inputs (e.g. pixels that are white for all digits). A fully-connected architecture should rather be applied to a selection of classification and regression tasks where at least some of the input variables are mutually independent. Why not take a random pick from: https://github.com/EpistasisLab/pmlb

**Hyperparameters:** It seems that the same learning rate was used for both the proposed architecture and the standard MLP. This seems inappropriate: log-likelihood (cross-entropy) loss yields higher gradients than the mean squared loss, as such one usually picks smaller learning rate value for cross-entropy. The comparison between architectures will only be fair if the hyperparameters are optimised separately for each, the proposed model and the MLP.

 **Formatting:** The authors do not always adhere to correct formatting standards. A few grammatical mistakes are present in the paper. Please see below for the suggested list of corrections:

“disconnected from biological reality. (Akomolafe, 2013).” - please remove the full stop prior to the reference.

“model depicted a noticeable increase” - model exhibited

“While Hopfield use” - While Hopfield used

“UNet++(Zhou et al., 2018), “ - add a space in front of the bracket

“Though the forward propagation mechanism of the proposed model echoes the forward propagation mechanism of conventional RNNs, in conventional RNNs, the activations of the hidden neurons, obtained from the prior time step, are used to calculate the output of the hidden layer, whereas in CHNNet, the current pre-activations of the hidden neurons are used to calculate the output of the hidden layer.” - this sentence is too long, please break it up into three separate sentences.

“equation 3 and equation 4” - Equation (3) and Equation (4)

I suggest that the contributions are listed in present tense.

**Questions:**

*“the difference between the cost of CHNNet, generated at two sequential time steps, is greater than that of the conventional FNN; that is, CHNNet converges faster than the conventional FNN.”* — is this really a valid statement? From the loss landscape perspective, a higher difference in loss values implies “steeper” gradients. Whether or not that would lead to faster convergence is not easily answered, in my opinion. With very steep gradients one might experience higher oscillations and stronger sensitivity to the learning rate value. I think the authors should only state what they know for certain: that the proposed approach provably yields higher gradients (which may lead to faster convergence).

The p-values show that the difference in performance is not statistically significant for the proposed model and the MLP. What about convergence speed? Can that be quantified and statistically evaluated for significance? Also, what about runtime efficiency—in absolute terms, when the number of parameters is comparable, which architecture is quicker to train? Please evaluate/comment on this.

---

> ### Author Response · Authors · 2023-11-21
>
> We are truly grateful to the reviewer for the highly detailed and constructive feedback on our paper. Below, we address each question and concern in a point-by-point fashion:
>
> #### **Weaknesses:**
> - Benchmarking experiments: We had conducted additional experiments on the Abalone, Iris, Boston Housing, and Breast Cancer datasets and got similar results to those shown in the paper. However, due to page restrictions, we could not add the results to the paper.
>
> - Hyperparameters: Regarding the hyperparameters, our hypothesis was that because of the set of weights referred to as $w_2$ in the paper, the model should outperform FNN in complex datasets. We had conducted additional experiments and concluded that CHNNet outperformed FNN in complex datasets like the FMNIST and EMNIST datasets.
>
> - Formatting: We will follow the suggested formatting in the camera-ready version of the paper if it gets accepted.
>
> #### **Questions:**
> - We will state the suggested statement about the steeper gradient in the camera-ready version of the paper if it gets accepted.
> - The p-values show that the difference in accuracy gained by the models is not statistically significant in many cases. However, our main goal was to get faster convergence than FNN, and in some experiments, we also got performance gains compared to FNN. Further, with an equal number of parameters in CHNNet and FNN, the number of computations per epoch is nearly equal as well. Thereby, the training time for CHNNet and FNN with an equal number of parameters will be nearly equal as well.

---

### Official Review · Reviewer_E1uY · 2023-10-27

**Soundness:** 1 poor
**Presentation:** 2 fair
**Contribution:** 2 fair
**Rating:** 3
**Confidence:** 4

**Summary:**

This paper proposes what is claimed to be a more robust model of artificial neural networks where the hidden neurons, residing in the same hidden layer, are inter-connected thus leading to an increase in convergence, when compared to feed-forward neural networks.

This is an interesting idea, inspired by the lateral connections omnipresent in the neural systems of primates, and in fact responsible for the center-surround motif in the retina. However, the mathematical formulation of the recurrence thus occurring in the networks seems incorrect, and equivalent to a feed-forward neural network. In more detail, the proposed formulation for a layer l is as follows:

    A(l) = f(Z(l))  where
        Z(l) =  W1(l) A(l-1) + W2(l) H(l) + B(l)
        H(l) = W1(l) A(l-1) + B(l)

where A(l) is the output of layer l, f is the activation function, and H(l) are the horizontal connections (Equations 1-3, on Page 3). However, by replacing H(l) with its definition one obtains:

    Z(l) =  (W1(l) + W2(l) W1(l)) A(l-1) + W2(l) B(l) + B(l)
          = W3(l) A(l-1) + B3(l)

Hence, the result is a classic feed forward computation unit of the form:

        A(l) = f(Z(l))  where
            Z(l) = W3(l) A(l-1) + B3(l)

Hence, by trying to avoid the recursion inherent in the lateral inhibition network, by taking the "hidden output" to be what the unit would generate as output in the absence of horizontal connections, leads to a network that is in all respects equivalent to a feed-forward network.

It might be the case that this partitioning of the computation leads in some cases to a faster convergence, but in this case the authors have to make a better case for the way in which the mathematically correct recurrent formulation is replaced with a feed-forward variant.

**Strengths:**

This paper introduces an interesting idea, inspired by the lateral connections omnipresent in the neural systems of primates, and in fact responsible for the center-surround motif in the retina.

**Weaknesses:**

However, the mathematical formulation of the recurrence thus occurring in the networks seems incorrect, and equivalent to a feed-forward neural network.

**Questions:**

Why have you avoided a recurrent formulation?

---

> ### Author Response · Authors · 2023-11-21
>
> We are truly grateful to the reviewer for the highly detailed and constructive feedback on our paper. Below, we address each question and concern in a point-by-point fashion:
>
> #### **Questions:**
> - The recurrent formulation will make the model a variant of RNN architectures. However, CHNNet is different from RNN, as discussed in section 1.1, and is in fact similar to a conventional FNN with intra-connected hidden layers. The novelty of the model lies in the introduction of a new set of weights, responsible for the rapid convergence as discussed in section 3.3 of the paper, and the approach used to connect the hidden neurons.

---

> > ### Comment · Reviewer_E1uY · 2023-11-22
> >
> > Thank you for the additional information and the clarifications. The paper can be considerably strengthen, by admitting right away that the composition of the linear transformations is in fact equivalent with another linear transformation, and pointing out (with extensive experiments) that nevertheless, the additional weights increase convergence and accuracy.

---

### Official Review · Reviewer_imnR · 2023-11-01

**Soundness:** 2 fair
**Presentation:** 2 fair
**Contribution:** 2 fair
**Rating:** 3
**Confidence:** 4

**Summary:**

This paper presents a neural architecture that interconnects hidden neurons on the same hidden layer.
It also presents the learning laws for updating the weights using backpropagation.
The central of this work is that the new architecture promotes rapid convergence.
It also presents a rapid convergence proof and some experimental results to support the claim.

**Strengths:**

The paper is well-organized

**Weaknesses:**

(1). It appears that the mathematical model for this new model reduces to the conventional feedforward network without intra-hidden layer connections.

 Equations (1) and (2) seem to suggest that  $H^{[l]}$ is a linear transformation of input $A{[l-1]}$. If you compose linear transformations, you get a linear transformation. Specifically, if you put equation (2) in equation(1), you will get a new linear transformation that is similar to using no intra-hidden layer connection.


(2). I think the experiments are insufficient:

The number of training epochs before convergence does not give the full picture. The claim of rapid convergence fails to account for the overhead per epoch due to the extra intra-layer computations. Also, I think the authors should compare this work with the use of skip connections.

:

**Questions:**

(1). Please can you provide more information on how this model architecture differs from the conventional feedforward network with an intra-connection layer?   Especially with the simplification of plugging equation (2) into equation (1)

(2). Can you provide more information about the compute time per epoch?

---

> ### Author Response · Authors · 2023-11-21
>
> We are truly grateful to the reviewer for the highly detailed and constructive feedback on our paper. Below, we address each question and concern in a point-by-point fashion:
>
> #### **Weaknesses:**
> - We omitted experiments with skip connections, as skip connections can be added to both CHNNets and conventional FNNs. As a result, skip connections can benefit both CHNNets and FNNs equally.
>
> #### **Questions:**
> - The proposed architecture is in fact a conventional FNN with intra-connected hidden layers. The novelty of the model lies in the introduction of a new set of weights, responsible for the rapid convergence as discussed in section 3.3 of the paper, and the approach used to connect the hidden neurons.
> - With an equal number of parameters in CHNNet and FNN, the number of computations per epoch is nearly equal as well. Additionally, in section 4.4, we conducted experiments with CHNNet and FNN having nearly equal numbers of parameters and showed that CHNNet outperformed FNN in such setups as well.

---

### Official Review · Reviewer_Qbio · 2023-11-08

**Soundness:** 3 good
**Presentation:** 3 good
**Contribution:** 3 good
**Rating:** 6
**Confidence:** 3

**Summary:**

This paper proposes a novel FFNN architecture where neurons in hidden layers are "connected to themselves" (albeit without a non-linearity), as well as being connected to previous and future layers. The authors demonstrate that this architecture leads to faster convergence and better final performance on a set of MNIST tasks. They also acknowledged the potential comparison issue that self-connections result in more parameters, and performed another raft of experiments which normalized for parameter count between traditional and CHHNet architectures, and showed continued increased convergence for CHHNets even in that setting.

**Strengths:**

- The paper was straightforward in its proposal and the evidence for it, and was thus easy to engage with as an isolated idea
- I appreciated the authors pre-empting the concern about parameter-count-parity; I think I would indeed have found these results less compelling in the absence of that experiment
- I appreciated the ways the authors drew comparisons with and also distinctions from other self-loop architectures like RNNs

**Weaknesses:**

- The limited number of datasets tested on strikes me as the primary experimental weakness of this paper: given the similarities between all of the MNIST and MNIST-adjacent datasets, it's a little hard to tell to what extent these results suggest a more general conclusion, vs just a property of training FFNNs on MNIST and similarly-structured datasets.
- I think the convergence speed proof could have done with a little more explanation to make the intuitions of the steps in the chain more clear

**Questions:**

- This is more a question than a suggestion, but given the central role MLPs are currently playing in modern transformer architectures, it would be interesting to see whether this approach leads to better performance/faster convergence in that setting (i.e. as a way to modify the parameterization of internal MLPs)

---

> ### Author Response · Authors · 2023-11-21
>
> We are truly grateful to the reviewer for the highly detailed and constructive feedback on our paper. Below, we address each question and concern in a point-by-point fashion:
>
> #### **Weaknesses:**
> - We had conducted additional experiments on the Abalone, Iris, Boston Housing, and Breast Cancer datasets and got similar results to those shown in the paper. However, due to page restrictions, we could not add the results to the paper.
> - We will add more explanation to the convergence proof in the camera-ready version of the paper if it gets accepted.
>
> #### **Questions:**
> - We appreciate your suggestion and will work with modern transformer architectures in the future.

---

### Meta-Review · Area_Chair_kB9K · 2023-12-06

**Metareview:**

This paper presents a neural architecture that interconnects hidden neurons on the same hidden layer. It is argued that learning parameters of this architecture can happen faster the training a conventional architecture. While ideas of this kind are very interesting for machine learning, partly due to their prevalent presence in the neural systems of primates as stated by reviewer E1uY, the draft can benefit from a major revision. The current presentation has caused some confusion about whether the proposed model is related to a recurrent or feedforward model as pointed out by reviewers imnR and E1uY. In their response, the authors stated that the proposed architecture is indeed related to a feedforward model (but has a different training dynamics, which may speed up training). However, E1uY still thinks the paper can be made stronger by clearly admitting in the main paper that the proposed architecture is closely related to a conventional feedforward model, and pointing out (with extensive experiments) that nevertheless, the additional weights increase convergence and accuracy. Another major concern across the board was limited experimentation. The authors responded that they had conducted additional experiments on the other datasets but due to page restrictions, could not add these results to the paper. I believe the submission can definitely benefit from additional experiments, and if the authors run into space issue, they can use supplementary appendix to include their additional results. Finally, reviewer  Qbio believes presentation of the convergence proof can improve by a more detailed explanation.

In summary, the paper is studying an interesting problem, but in its current state is not ready for publication. I encourage the authors to revise their submission based on the provided feedback and resubmit.

**Justification For Why Not Higher Score:**

The paper needs major improvement on its empirical assessment and clarity of presentation

**Justification For Why Not Lower Score:**

N/A

---

### Decision · Program_Chairs · 2024-01-16

Reject